# The Current State of Chromatin Immunoprecipitation (ChIP) from FFPE Tissues

**DOI:** 10.3390/ijms23031103

**Published:** 2022-01-20

**Authors:** Stefano Amatori, Mirco Fanelli

**Affiliations:** Molecular Pathology Laboratory “PaoLa”, Department of Biomolecular Sciences, University of Urbino Carlo Bo, Via Arco d’Augusto 2, 61032 Fano, PU, Italy

**Keywords:** FFPE tissues, archival samples, chromatin, chromatin immunoprecipitation (ChIP), cancer epigenetics

## Abstract

Cancer cells accumulate epigenomic aberrations that contribute to cancer initiation and progression by altering both the genomic stability and the expression of genes. The awareness of such alterations could improve our understanding of cancer dynamics and the identification of new therapeutic strategies and biomarkers to refine tumor classification and treatment. Formalin fixation and paraffin embedding (FFPE) is the gold standard to preserve both tissue integrity and organization, and, in the last decades, a huge number of biological samples have been archived all over the world following this procedure. Recently, new chromatin immunoprecipitation (ChIP) techniques have been developed to allow the analysis of histone post-translational modifications (PTMs) and transcription factor (TF) distribution in FFPE tissues. The application of ChIP to genome-wide chromatin studies using real archival samples represents an unprecedented opportunity to conduct retrospective clinical studies thanks to the possibility of accessing large cohorts of samples and their associated diagnostic records. However, although recent attempts to standardize have been made, fixation and storage conditions of clinical specimens are still extremely variable and can affect the success of chromatin studies. The procedures introduced in the last few years dealt with this problem proponing successful strategies to obtain high-resolution ChIP profiles from FFPE archival samples. In this review, we compare the different FFPE-ChIP techniques, highlighting their strengths, limitations, common features, and peculiarities, as well as pitfalls and caveats related to ChIP studies in FFPE samples, in order to facilitate their application.

## 1. Introduction

Chromatin in eukaryotes is a finely organized nuclear complex of genomic DNA, histones, and non-histone proteins. The lowest level of chromatin organization is represented by nucleosomes, which consist of 147 base pairs of DNA wrapped ~1.7 turns around a histone octamer core. The octamer is composed of four heterodimers of the core histone proteins H2A, H2B, H3, and H4, while a fifth histone, histone H1, binds to internucleosomal DNA to stabilize higher-order structures. Chromatin function is mainly regulated by histone post-translational modifications (PTMs), which consist of enzyme-mediated chemical modifications of specific histone residues, among which those targeting the N-terminal tail of histones seem to play a major regulatory role. Different ATP-dependent chromatin remodelers are guided by these modifications to control the chromatin state and regulate the gene expression by making chromatin accessible or not to transcriptional regulatory complexes. Chromatin structure indeed has a crucial role in various processes including activating or repressing transcription to control functions such as the cell cycle, DNA damage repair, and cell fate [1].

Chromatin structure misregulation was found to play a main role in several human diseases, including cancer. Many histone-modifying enzymes and chromatin remodeling complexes malfunction in cancer, and their alteration is believed to be a key mechanism in tumor development and progression [2,3]. Aberrant expression and/or incorporation of histone variants have also been linked to cancer and to more aggressive cancer phenotypes [4,5]. In addition, recent studies demonstrated that some histone genes are frequently mutated in cancer, and many investigations to sustain their role in tumor transformation have been accumulated [6,7,8].

Chromatin immunoprecipitation (ChIP) has strongly enhanced our knowledge about the meaning of protein–DNA interactions, as well as the significance of histone PTMs in many different biological contexts. ChIP consists of the isolation of chromatin fragments from a biological matrix and the consequent immunoselection of a protein of interest to identify the genomic loci associated with it [9,10]. In this technique, the starting material (cultured cells of fresh/frozen tissues) is normally fixed by formaldehyde, and chromatin is extracted and fragmented by controlled sonication. Chromatin is then subjected to immunoselection, using an antibody directed against a specific chromatin protein, and the immunoselected chromatin is then decrosslinked, while the DNA is purified and, finally, studied at single-locus or genome-wide levels. ChIP can be used to study transcription factor (TF) binding but has attracted great attention for the study of histone PTMs after the surge of interest in epigenetic research [11,12]. The advent of next-generation sequencing (NGS) has expanded the potential of ChIP, opening the doors to the detailed mapping of histone PTMs and TF binding sites over the entire genome [13,14].

For many years, ChIP has been mainly applied in studies through which the epigenetic features of cancer have been investigated using cell lines that, unfortunately, are often a poor model to investigate chromatin dynamics across cancer progression and, most importantly, are subjected to epigenetic changes as a consequence of the adaptation to culture conditions [15,16]. The investigation of cancer epigenetics in primary tumors is the means to overcome these limitations. Formalin fixation followed by paraffin embedding (FFPE) is the standard method for long-term preservation of most archived pathological specimens. These samples indeed allow the pathological evaluation of tissue histology by immunohistochemistry (IHC) and immunofluorescence (IF), while maintaining the possibility to isolate not only high-quality DNA but also RNA [17,18,19,20,21].

In summary, the benefits of using FFPE include (i) the access to large tissue archives, (ii) the analysis of real pathology samples, (iii) the possibility to isolate specific cellular populations in the tissue, and, most importantly, (iv) the access to patients’ clinical history, opening the way to retrospective studies and, thus, to the discovery of new potential clinical markers.

In 2010, our research group was the first to propose a technique that allows epigenomic studies in FFPE tissues [22]. From that date on, several other techniques (including an upgrade of our first procedure) have been developed with the intent of improving chromatin profiling in FFPE tissues, underlining the importance of demonstrating the applicability of the ChIP procedure by using FFPE samples, paving the way to the development of this new area of research [23,24,25,26,27,28]. In particular, the main hurdle to overcome is the difficulty to reproducibly extract high-quality chromatin from FFPE samples, mainly as a result of the extended and variable level of formalin fixation, as well as the different conditions and timing of sample storage.

In this review, we aim to examine the ways through which different research groups and companies have dealt with chromatin extraction and immunoprecipitation from FFPE tissues, while also providing recommendations/guidelines and practical examples to improve the success of chromatin studies in such intriguing samples.

## 2. The Development of FFPE-ChIP Techniques over Time

There are two main ways through which a tissue specimen can be stored: flash freezing (or snap freezing) in liquid nitrogen, followed by storage at −80 °C, and fixation in formalin followed by paraffin embedding (FFPE). FFPE samples have attracted attention in the last years because they represent the standard way to preserve patient samples for diagnostic purposes. The first technique that allowed the study of histone marks and transcription factors in FFPE specimens was developed by our research group and published in 2010 (Figure 1 and Table 1) [22,29].

In this procedure, which we called pathology tissue chromatin immunoprecipitation (PAT-ChIP), chromatin isolation from FFPE samples is achieved by both physical extraction, using a probe sonicator at high amplitude, and controlled micrococcal nuclease (MNase) digestion. We demonstrated that FFPE-isolated chromatin can be immunoselected, applying the procedure to both genome-wide (coupled with NGS) and to locus-specific (i.e., real-time qPCR) studies using different tissues such as seminoma, breast cancer, and lung carcinomas [22,30]. In 2016, a further procedure named fixed-tissue chromatin immunoprecipitation sequencing (FiT-seq) was proposed by Cejas and colleagues (Figure 1 and Table 1) [23]. This technique, which is essentially based on a controlled proteinase K digestion and chromatin shearing by sonication through a focused ultrasonication system, was successfully applied to different archival human samples (seminoma, breast cancer, bladder cancer, and colorectal cancer). However, as subsequently stated by the same authors, this new technique fails to resolve H3K27ac, suggesting a possible degradation of H3K27ac due to proteinase K activity that may potentially also affect other kinds of epitope in certain experimental conditions [26]. In the meantime, we found that genome-wide studies from FFPE archival samples by PAT-ChIP may be limited by the low efficiency of chromatin isolation often due to extensive tissue fixation introduced during routine pathological processing [24]. Thus, to overcome the issue of FFPE over-fixation we improved the procedure by introducing, for the first time in this type of experimental approach, a heat-mediated limited reversal of crosslinking (1 h at 80 °C) to partially revert the effects of extensive formalin fixation. The new technique was named enhanced PAT-ChIP (EPAT-ChIP) and was found to improve the success of genome-wide studies of both histone methylation and acetylation in archival samples (invasive breast carcinoma) [24]. Other authors suggested that this technique suffers from promoter bias when applied to the study of H3K27ac [26]. However, this seems not to be the case, since genomic annotations of H3K27ac obtained by EPAT-ChIP were found to overlap with those produced by other research groups (e.g., ChIP-seq from ENCODE) [24]. To date, PAT-ChIP has been applied by different research groups to locus-specific and genome-wide studies, supporting the reproducibility of the entire procedure [31,32,33,34,35,36,37,38,39,40,41,42,43].

Notably, the last 2 years have seen the birth of three other procedures. In 2019, Zhong and colleagues proposed chromatin extraction from FFPE tissues (Chrom-EX PE), a technique that exploits heat (16 h at 65 °C) to extract high-quality chromatin through a focused ultrasonicator [25]. Unfortunately, they applied their technique to genome-wide studies only in livers and spleens from mice expressly produced for their study, and the applicability to real FFPE archival tissues remains to be rigorously proved. One year later, the developers of FiT-seq published the detailed protocol of a modified version of their procedure, named FiTAc-seq (Figure 1 and Table 1), which, unlike their previous one, can be used to map H3K27ac in archival samples (neuroendocrine tumors, bladder cancer, breast cancer, brain metastasis, seminoma, and melanoma) [26]. The procedure makes use of heat at the same temperature and duration of incubation of the method proposed by Zhong and colleagues (16 h at 65 °C) and was applied by the same authors in a study published the year before (even if the details of the method are not described there) [44].

A further procedure, named RCRA ChIP-seq, was published a few months ago by Kaneko and colleagues (Figure 1 and Table 1) [27]; again, a heating step (1 h at 65 °C followed by 30 min at 90 °C) was proposed to efficiently extract, through a focused ultrasonicator, high-quality chromatin from archival lung adenocarcinoma samples. Interestingly, the procedure was applied through a dual-arm robot to 69 samples, being the first attempt to automate ChIP-seq studies from FFPE samples. In the meantime, two commercial kits have also been developed. A first version, named ChIP-IT FFPE Chromatin Preparation (Active Motif, Carlsbad, CA, USA, cat #53030), was marketed by Active Motif from 2014 and joined in 2018 by a second version (ChIP-IT FFPE Chromatin Preparation II, Active Motif, cat #53031—Figure 1 and Table 1). The two versions are quite different, with the second one showing a high reduction of complexity with respect to the first and the introduction of a heating step even if at very low temperatures (1 h at 50 °C). We found four studies citing the usage of this kit which is, together with PAT-ChIP, the only procedure applied by research groups other than their own developers [25,45,46,47]. Unfortunately, one of these groups stated that the kit (ChIP-IT FFPE Chromatin Preparation II) did not work [25]. A second kit named truChIP FFPE (truChIP FFPE Chromatin Shearing Kit, Covaris Cat. #520257) was developed by Covaris and marketed in 2018 (Figure 1 and Table 1). This is the only procedure in which chemical deparaffination and the rehydration of tissues are omitted. This kit uses, similarly to FiT-seq, proteinase K digestion of proteins to facilitate the extraction of chromatin, while paraffin is emulsified by focused ultrasonication. The usage of this kit has never been cited in the literature. Taken together, the procedures introduced to date suggest that controlled reversal of crosslinking using high temperatures may be the key to the success of chromatin studies in FFPE archival samples. A schematic representation of the main steps shared by the different procedures described above is reported in Figure 2.

A few months ago, a further technique that combines a novel fusion protein of hyperactive Tn5 transposase and protein A (pA−Tn5) transposition and T7 in vitro transcription to perform chromatin studies in FFPE tissues was proposed (Figure 1 and Table 1) [24]. This procedure is a first attempt to apply the principles of novel strategies for chromatin profiling already used for fresh/frozen samples, such as CUT&RUN [48] and CUT&Tag [49], to archival samples with the intent of reducing the quantity of starting material.

In the following sections, we aim to describe in detail and compare the different experimental strategies used to achieve chromatin profiling in FFPE samples, from the preliminary evaluation of the specimens to chromatin extraction and immunoselection, including the final quality tests that might precede the NGS sequencing.

### 2.1. Preliminary Evaluations

Tumor samples are commonly characterized by the presence of variable amounts of normal tissue that hinders the correct identification of the cellular population contributing to a specific phenomenon and decreases the reliability of final data. Thus, a histological analysis by hematoxylin and eosin staining is strongly recommended, while also considering its compatibility, already demonstrated, with chromatin extraction and immunoselection [30]. When the tissue component of interest is not homogeneous, tissue sections can be macrodissected or subjected to laser capture microdissection (LCM). Most procedures make use of slides of 5–10 µm thickness, and they have been demonstrated to also be compatible with LCM [23,24,26,27,29]. Histological analysis plays an important role in the success of a study since high and constant purity allows obtaining robust data for final computational analysis.

As already stated, genome-wide studies from FFPE archival samples may be hindered by the low efficiency of chromatin isolation, often due to extensive tissue fixation introduced during routine pathological processing. A solution containing 3.7–4% of formaldehyde (FA) is routinely used as fixative reagent with a length of fixation that is influenced by different factors (e.g., day of tissue resection, operators/instrument availability, etc.) [50]. Despite the recent advancements in the standardization of FFPE tissue preparation, the fixation times are extremely variable [51,52,53]. Time between surgery and tissue storage can also vary and may impact the preservation of epitopes, thus introducing significant unwanted bias.

In addition, although preventing hydrolysis during prolonged storage minimizes protein degradation, FFPE tissues deteriorate over time [54]. Thus, time and storage conditions can further contribute to the variability of the starting material.

Given this evidence, it is suggested not to enroll, when possible, over-fixed samples (e.g., samples fixed for more than 72 h) and samples stored for a long time to maximize the chances of extracting large quantities of high-quality chromatin.

### 2.2. Chromatin Extraction from FFPE Tissues

Before starting to analyze how the different techniques deal with chromatin extraction from FFPE specimens, some additional considerations must be taken into account. Independently from formalin fixation and paraffin embedding, the type of tissue is per se an important source of variation. Cellular density and structural features of each tissue may indeed impact chromatin extraction efficiency. Thus, it is strongly suggested to adjust sonication conditions in function of the tissue processed and to test different conditions before starting a new study.

For example, chromatin fragmentation should be checked by running decrosslinked and purified DNA on an agarose gel or through automated capillary electrophoresis prior to immunoselection. Best results are obtained when most of the fragments range between 200 and 1000 bp, since lower-size fragments normally indicate high degradation of the sample (due to excessive sonication and/or poor quality of the starting material), while higher-size fragments are normally lost during library preparation and could reduce the final resolution of the assay. Many of the FFPE-ChIP procedures herein described agree to evaluate the fragmentation of the extracted chromatin before proceeding with the immunoselection [22,24,25,27].

The amount of starting material can also influence chromatin extraction by limiting the effect of sonication. Thus, chromatin should be extracted starting from equal amounts of starting material to reduce possible bias deriving from extraction-dependent chromatin selection.

With the intent of deciphering which procedural parts related to chromatin extraction from FFPE are most relevant, we conducted a comparative analysis of the different procedures proposed in these years. The thickness of the sections processed varies from 5 to 20 µm, while the number of sections required is normally quite low, ranging from 1 to 4, with FiT-seq being the only one suggesting a higher number (10 whole sections—Table 1). However, we found that the amount of starting material needed is not written in stone and must be adjusted as a function of the kind of tissue, the size of the sample, and the eventual execution of macro- or microdissection. Even in this case, a preliminary setting must be performed before starting a new project.

Regarding the initial processing of samples, all the procedures agree to perform paraffin removal and rehydration of tissue sections (Table 2).

Deparaffination is achieved with a few passages in a deparaffination agent, such as xylene (FiT-seq, Chrom-EX PE, FiTAc-seq, RCRA ChIP-seq) or limonene-based agents (PAT-ChIP and EPAT-ChIP). Rehydration is also quite standardized, with minor variations in passages in decreasing alcohol concentrations, from 100% to 0% (Table 2). The only exception is represented by the commercial kit truChIP FFPE from Covaris, which proposes an interesting focused ultrasonication-mediated deparaffination (Table 2). After paraffin removal and rehydration, all the procedures involve further processing of the sample prior to chromatin extraction by sonication, although, in this case, the solutions proposed are more variable. In PAT-ChIP and EPAT-ChIP, for example, sections are lysed in the presence of a nonionic detergent and subjected to fragmentation by sonication at low amplitudes to prepare the sample for subsequent processing (Table 2).

Concerning the issue of an extensive fixation of the tissue, introduced during routine pathological processing [24], the FFPE-ChIP procedures are characterized by different strategies to overcome this problem. Excessive sonication indeed may alter the antigenic properties of chromatin leading to loss of antigenicity and must be avoided. For example, initial techniques, such as PAT-ChIP and FiT-seq, proposed controlled chromatin digestion to reduce the complexity of the tissue facilitating the extraction of chromatin. The original PAT-ChIP procedure indeed used MNase to partially digest DNA [22], while FiT-seq makes use of proteinase K with the intent of, as stated by the authors themselves, “helping to resolve the excessive number of crosslinks introduced by long exposure to formaldehyde” (Table 3) [23].

However, while DNA digestion is quite easy to control through a simple run on an agarose gel, protein digestion may be “dangerous” considering that a protein is just the target of the assay. The failure in the application of the FiT-seq by our research group and the statement by the authors themselves that FiT-seq fails to work with H3K27ac may indicate the necessity of a tricky fine-tuning of the technique [26].

The main step forward in chromatin extraction from FFPE tissues is probably the introduction of a heating step to partially revert the crosslinking of FFPE tissues. The first technique to develop such strategy was EPAT-ChIP in late 2018 (Table 3) [24], followed by all the subsequently published procedures, such as Chrom-EX PE, FiTAc-seq, and RCRA ChIP-seq (Table 3) [25,26,27]. However, these four procedures differ in temperature, time of incubation, and the presence of detergents. Summarizing, while both Chrom-EX PE and FiTAc-seq indicate an incubation of 16 h at 65 °C, EPAT-ChIP and RCRA ChIP-seq suggest shorter times and higher temperatures (1 h at 80 °C and 1 h at 65 °C followed by 30 min at 90 °C, respectively) reducing by 1 day the duration of the procedure (Table 3). This can be an important advantage, especially considering the high time consumption that characterizes all these procedures.

A final and important point to discuss regards the conditions of sonication. In addition to the different types and concentrations of detergents used to achieve extraction and solubilization of chromatin, the main distinction found is between the usage of probe sonicators or focused ultrasonicators. PAT-ChIP and EPAT-ChIP, together with the procedure described for the commercial kit ChIP-IT FFPE Chromatin Preparation II from Active Motif, are the only procedures that involve the use of a “canonical” probe sonicator (Table 3) [22,24]. All the other techniques make use of focused ultrasonication systems such as the one from Covaris (FiT-seq, FiTAc-seq and the commercial kit truChIP FFPE—Table 3) [23,26] or Bioruptor from Diagenode (Chrom-EX PE and RCRA ChIP-seq—Table 3) [25,27]. Although focused ultrasonication offers advantages in terms of reproducibility and recovery, probe sonicators are much more widely distributed, allowing the execution of the technique in all standard laboratories.

### 2.3. Chromatin Immunoselection

Once the objective of getting high amounts of good-quality chromatin is achieved, the criticisms related to chromatin immunoselection are largely comparable to those of standard ChIP techniques. Incubation conditions are influenced by the ratio between the amount of chromatin and antibody, the volume of incubation, the presence of detergents and salts, and the time and the temperature of incubation. All these aspects, together with the system used to capture the chromatin–antibody complexes and the washes used to eliminate the not specific binding of chromatin, must be carefully gauged. Each ChIP technique, regardless of the starting material, makes use of its own strategy to reach the final goal, which is to maximize the binding of the target epitopes, while reducing as much as possible the noise caused by unwanted nonspecific binding of chromatin.

Regarding the chromatin required for each immunoselection (input), many of the techniques examined suggest to decrosslink, purify, and quantify the DNA from an aliquot of the extracted chromatin to precisely estimate its amount [24,25,26]. This step increases by 1 day the completion of the procedure but ensures the performance of the assay with both controlled fragmentation and amount of chromatin. When declared, all the procedures agree to submit to immunoselection a quantity of input between 0.4 and 4 µg (Table 4).

Interestingly, FiTAc-seq makes use of a carrier-assisted ChIP method developed by other authors [55] that entails the mixing of input chromatin with large quantities of recombinant histone H2B and total RNA to improve the biochemical conditions of incubation with the antibody [26], an approach similar to the Bryan Turner lab-pioneered Carrier-ChIP (CChIP) method [56]. However, although useful, this strategy involves the use of high amounts of immunoglobulins, which can have a significant impact on the final cost of a project.

Although the composition of the incubation mixture can vary a lot, almost all the procedures perform the incubation with the antibody for 16 h at +4 °C, with the intent of optimizing the interaction between the antibody and its target epitopes (Table 4). However, RCRA ChIP-seq is the only technique among those herein analyzed that uses short times of incubation (40 s) exploiting an ultrasonic water bath (Table 4) [27].

Chromatin–antibody complexes are normally captured using protein G and/or A beads that can be agarose or sepharose beads and magnetic or not magnetic (Table 4). Column systems can also be used, as reported in FiT-seq technique (Table 4) [23]. Elution is performed using solutions containing 1% SDS, while chromatin is fully decrosslinked by a 16 h incubation at 65 °C, often in the presence of 0.2 M NaCl, and proteins are eliminated by proteinase K digestion (Table 4). Except for PAT-ChIP, all the FFPE-ChIP procedures, including EPAT-ChIP, exploit fast column-based kits for the final purification of DNA (Table 4).

As for all ChIP protocols, the quality of the antibody is fundamental for the feasibility of the entire procedure. For this reason, it is strongly suggested to get, when available, ChIP- or ChIP-seq-grade antibodies. However, based on our direct experience, the single lots of antibodies may not be tested by suppliers and the observation of high lot-to-lot variation is not to be considered a remote event. Thus, the preliminary test of different lots of antibodies is fundamental to ensure a homogeneous experimental condition. In addition, coupling this test with the real-time qPCR analysis of regions known to be enriched or not enriched by the histone PTM under study can provide an idea of the enrichment that can be reached with a specific immunoglobulin (see next section for details). The best condition will be the one showing the higher level of specificity of the immunoselection (the signal/noise ratio), while getting sufficient final DNA to produce a high-quality library (the optimal amount of DNA can vary as a function of reagents or the kit used to prepare the libraries). Details on the strategies to evaluate the specificity of the immunoselection are described in the next section.

### 2.4. DNA Analysis and Experimental Controls

Once final DNA (bound) is obtained, it is strongly suggested to perform a few quality checks before proceeding with library preparation and sequencing. DNA is normally fluorometrically quantified, and the enrichment of total DNA, with respect to the input, can be estimated. DNA quantitation is important to ensure library preparation under controlled and comparable conditions, especially if several samples must be compared, limiting library and sequencing bias that can negatively impact the final data. As already stated before, some authors indicate that an evaluation of the specificity of the immunoselection should be performed prior to library preparation [24,25]. To reach this aim, primer pairs that allow the amplification of genomic regions known to be enriched and not enriched by the histone modification or TF of interest must be available for a real-time qPCR analysis. The regions of interest can be identified from previously published papers or exploring ChIP-seq databases such as the one from UCSC Genome Browser. An example of this kind of evaluation based on H3K4me3 ChIP-Seq data from the ENCODE project is shown in Figure 3.

Primer pairs for the analysis of histone PTMs such as H3K4me3, H3K27me3, H3K27ac, H3ac, H4ac, RNA polymerase II, and CTCF in human and murine models are already available in literature [22,24,29,30]. In addition, this analysis can be useful to control library preparation and sequencing, since the level of enrichment identified by qPCR is expected to be maintained after sequencing. However, the evaluation of the level of specificity of the immunoselection through this analysis is not always easy to interpret, since a cutoff value for the fold enrichment between the regions expected to be enriched and the one expected not to be enriched (which should represent the noise or background) for a specific antibody and sample is normally difficult to set. The titration of the antibody, already described in the previous section, can also contribute to get an idea of the level of enrichment that can be reached with a specific immunoglobulin.

To ensure control over the reproducibility of the assays performed at different times, a technical control of immunoselection might also be considered. For example, chromatin extracted from a cell line routinely used in the lab and already immunoselected with the same antibody can be used to this end. Lastly, negative controls, such as no antibody (mock) or an unrelated antibody (e.g., an immunoglobulin of the same class but not able to recognize chromatin proteins), can also be added to get an idea of the nonspecific binding of chromatin. However, it should be remembered that both these controls have limitations that can give a wrong idea of the specificity of the immunoselection; thus, the evaluation of enrichment over background described above should be preferred.

### 2.5. A Novel Enzyme-Tethering Strategy

Recently, enzyme-tethering strategies, such as CUT&RUN [48] and CUT&Tag [49], have become quite popular alternatives to canonical ChIP-seq procedures. These methods were developed to map in situ histone modifications and TF binding using a limited number of cells, while also allowing single-cell studies. In these techniques, chromatin is not extracted, but the enzymatic activity of MNase or hyperactive Tn5 transposase, guided by specific antibodies directed against the target under study, are exploited to isolate selected DNA. In particular, CUT&Tag uses a Tn5 transposase-based tagmentation to fragment chromatin while simultaneously introducing barcoded adapters for PCR amplification within intact cells [49].

A few months ago, a further technique exploiting an enzyme-tethering strategy for chromatin profiling in FFPE samples was published [28]. This technique is named FACT-seq and combines a novel fusion protein of hyperactive Tn5 transposase and protein A transposition and T7 in vitro transcription (IVT) to perform chromatin studies in FFPE tissues. Interestingly, the procedure was developed to work with nuclei isolated after enzymatic (collagenase and hyaluronidase treatment) and physical (syringe needle passing) disruption of the FFPE tissue. Nuclei are then counted and subjected to partial crosslinking reversal by heat before being incubated with the antibody directed against the target under study [28].

Before setting their procedure, the authors demonstrated that the standard CUT&Tag protocol is not suitable for FFPE samples, suggesting that fragmentation of DNA induced during crosslinking reversal of archival FFPE chromatin may be the cause of such failure. Thus, they thought to overcome this problem using pA-Tn5 loaded with an adapter containing the T7 promoter sequence. They hypothesized that, even if there are DNA breaks in the middle of DNA fragments, they could still decode the insertion sites of pA–Tn5 transposase by transferring broken DNA fragments into RNA molecules with IVT. The IVT RNA is then reverse-transcribed in cDNA, purified, and converted to double-stranded cDNA; then, the library is prepared using, again, Tn5 tagmentation [28].

Through their work, the authors reached the objective of being the first to develop a technique based on enzyme-tethering strategies, which allows the epigenomic profiling of a low number of nuclei (about 4000) from FFPE tissues. Unfortunately, unlike the standard CUT&Tag procedure, the protocol is time-consuming (5 days of work, excluding sequencing) and is based on several steps requiring enzymatic activities that may increase the experimental variability, with the need of a complicated fine-tuning of the technique. As declared by the same authors, the technique must be adapted as a function of the histone modification investigated (in particular the antigen retrieval step) [28]. The FACT-seq procedure represents an extraordinary starting point, but further studies are needed to develop a technique sufficiently robust and less tricky to be widely applied.

## 3. Conclusions and Future Prospects

The chromatin regulatory landscape combines histone modifications, TF binding, and the function of genomic elements to determine the biological function of chromatin. The large-scale epigenomic studies from the Roadmap Epigenomics Mapping Consortium (http://www.roadmapepigenomics.org (accessed on 14 November 2021)) has generated detailed and accurate descriptions and classifications of the functional states of regulatory elements in human stem cells and primary ex vivo tissues [11,57,58,59,60,61,62]. The combination of these data with chromatin accessibility by ATAC-seq and gene expression profiles from RNA sequencing (RNA-seq) will allow obtaining a wide picture of the tissue-specific epigenomic landscape from a multidimensional perspective.

Chromatin is profoundly rearranged in cancer inducing the activation of oncogenes, as well as the inactivation of tumor-suppressor genes, which play a role in cancer progression and invasion. FFPE samples represent the gold standard for storage of pathology samples. After our first proposal of a ChIP technique applicable to FFPE tissues, several other procedures have been published in the last 5 years. Recent protocols share the introduction of a heat-mediated limited reversal of crosslinking that facilitates the subsequent extraction of chromatin by sonication, suggesting it as a useful way to overcome the problem related to FFPE sample fixation.

However, in some cases, clinical biopsies may have very limited material, leading to significant sensitivity problems. Enzyme-tethering strategies working in situ to produce high-quality chromatin profiles are currently available for fresh/frozen cells and tissues. A first attempt to apply such techniques to FFPE archival samples was recently developed, although there is, in our opinion, considerable scope for improvement.

Single-cell technology is another important frontier in chromatin studies. Tissues are complex mixtures of cells and, although strategies to enrich the content of a tissue component such as macro- and microdissection can be useful, the presence of several cellular components may normally impact the final resolution of the assay and gives an incomplete picture of the epigenomic cellular landscape. scRNAseq and scATACseq techniques are currently available, and a few ChIP techniques allowing single-cell analysis have also been introduced in recent years [48,49]. We believe that the day is not far off when single-cell approaches will be applied to FFPE tissues to reveal the signals associated with the different cell types.

## Figures and Tables

**Figure 1 ijms-23-01103-f001:**
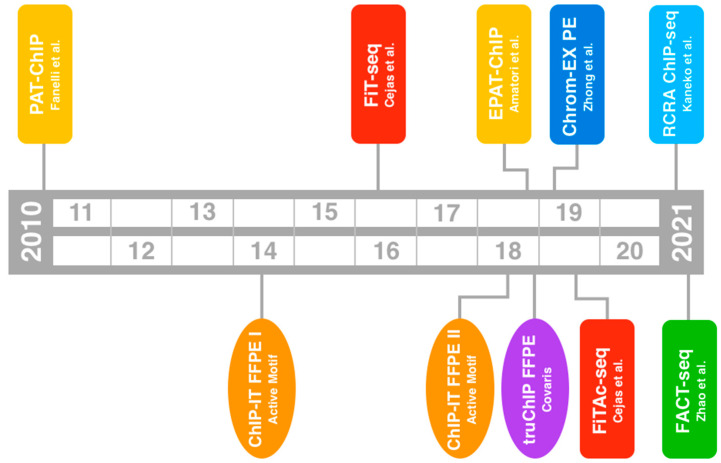
Timeline of the FFPE-ChIP procedures and kits introduced over time. Procedures are in rectangles while kits are in ovals. Specific colors are used for each research group and company.

**Figure 2 ijms-23-01103-f002:**
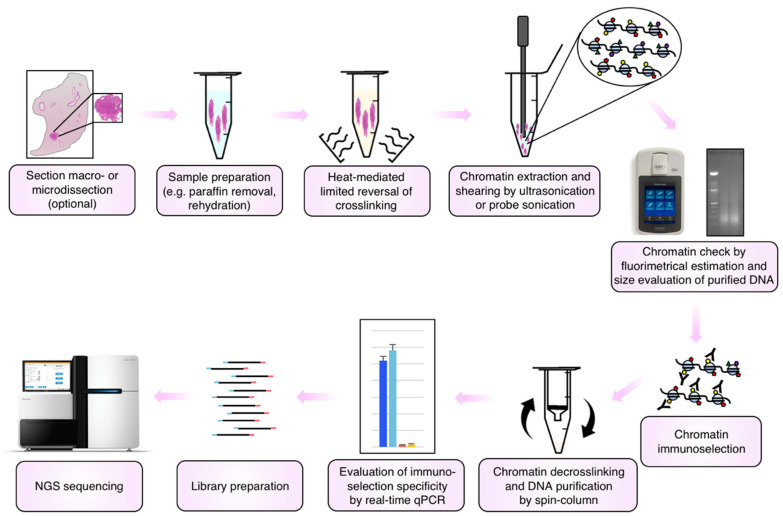
Schematic overview of the main steps shared by the different procedures described in this review. FFPE tissue sections can be collected from the block as a whole or dissected (macro- or micro-) to enrich the starting material of specific tissue components. The sections are deparaffinized and rehydrated prior to be subjected to limited reversal of crosslinking by heat and chromatin extraction and shearing by focused ultrasonication or canonical probe sonication. Then, chromatin should be checked prior to proceed with immunoselection: an aliquot of chromatin is taken and decrosslinked, and the DNA is purified, fluorimetrically quantified, and separated by electrophoresis to evaluate the size of fragments. Chromatin is then immunoselected using antibodies directed against the protein or histone modification of interest. After capture of chromatin–antibody complexes and their washing to remove the nonspecific chromatin binding, selected chromatin is decrosslinked and purified using spin-column strategies. Finally, a preliminary evaluation of the specificity of the immunoselection by real-time qPCR should be performed prior to library preparation and NGS sequencing.

**Figure 3 ijms-23-01103-f003:**
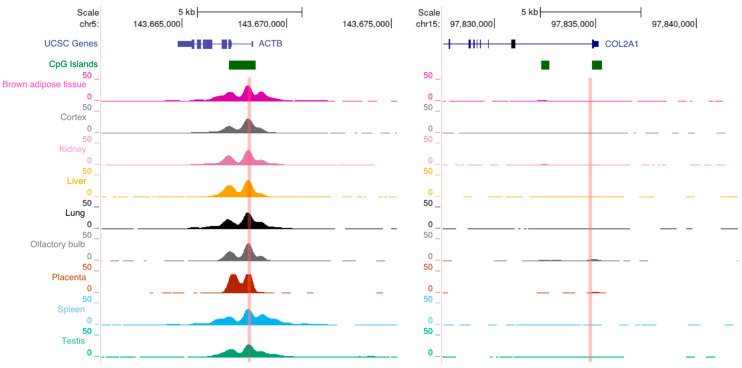
Identification of H3K4me3 control regions for the evaluation of the specificity of the immunoselection by real-time qPCR. Regions known to be enriched or not enriched can be used for antibody titration and to evaluate the specificity of the immunoselected DNA prior to library preparation and sequencing. Snapshots of mouse H3K4me3-enriched (*ACTB*) and not enriched (*COL2A1*) gene promoters are shown as example. ChIP-Seq data are from the ENCODE project and were taken from UCSC Genome Browser (http://genome.ucsc.edu (accessed on 7 November 2021)). The sequences we used to amplify as a control of specificity of H3K4me3 experiments on mouse tissues are indicated by the light-red vertical bars.

**Table 1 ijms-23-01103-t001:** Overview of the FFPE-ChIP procedures and kits analyzed in this review.

	PAT-ChIP	FiT-seq	EPAT-ChIP	Chrom-EX PE	FiTAc-seq	RCRA ChIP-seq	FACT-seq	ChIP-IT FFPE Chromatin Preparation II	truChIP FFPE
**Type**	Procedure	Procedure	Procedure	Procedure	Procedure	Procedure	Procedure	Commercial kit	Commercial kit
**Publication (or release) date**	11/2010	04/2016	11/2018	03/2019	07/2019	04/2021	09/2021	2018 (previous version: 2014)	12/2018
**Time needed (library prep. and seq. excluded) ***	4 days	N.A.	3 days	4 days	4 days	2–3 days	5 days	1 day (only chromatin preparation) **	1 day (only chromatin preparation) **
**Sections thickness (µm)**	10	10	10	20	10	8	10–20	5–10	10
**Number of starting sections**	4	10	2–4	2	2–4	1–2	1	up to 5	2
**References**	[22,29]	[23]	[24]	[25]	[26,44]	[27]	[28]		

N.A.: indicates that the information is not available in published manuscripts. * For all procedures, a further day may be required to evaluate chromatin abundance and fragmentation prior to immunoselection. ** The reported commercial kits include only chromatin extraction, while other kits of the same suppliers are indicated for chromatin immunoselection.

**Table 2 ijms-23-01103-t002:** Summary of the procedural steps used by the different FFPE-ChIP techniques and kits for tissue processing before chromatin isolation.

	PAT-ChIP	FiT-seq	EPAT-ChIP	Chrom-EX PE	FiTAc-seq	RCRA ChIP-seq	ChIP-IT FFPE Chromatin Preparation II	truChIP FFPE
**Deparaffination**	Histolemon, 5 times for 10 min	Xylene, 3 times	Histolemon, 5 times for 10 min	Xylene	Xylene	Xylene, 3 times	Xylene, 3 times	None
**Rehydration (%)**	100, 95, 70, 50, 20, 0	95, 80, 70, 50, 20, 0	100, 95, 70, 50, 20, 0	95, 70, 50, 20	100, 70, 0	100, 95, 80, 70, 50, 20, 0	100, 70, 50, 20, 0	None
**Lysis and RNA digestion**	30 min at RT with 0.5% Tween-20 and 10 µg/mL RNase A	1 h at 40 °C with 0.1% SDS	30 min at RT with 0.5% Tween-20 and 10 µg/mL RNase A	None	None	None	None	None
**Sonication (sample homogenization)**	Probe sonicator, 3 × 30 s at 40% amplitude	None	Probe sonicator, 3 × 30 s at 40% amplitude	None	None	None	None	None

Conditions are reported only if published or described in kit datasheets. Buffer composition: in the absence of other relevant features, only the concentrations of detergents and salts are reported.

**Table 3 ijms-23-01103-t003:** Summary of the procedural steps used by the different FFPE-ChIP techniques and kits for chromatin extraction.

	PAT-ChIP	FiT-seq	EPAT-ChIP	Chrom-EX PE	FiTAc-seq	RCRA ChIP-seq	ChIP-IT FFPE Chromatin Preparation II	truChIP FFPE
**MNase digestion**	1 min at 37 °C with 0.1 U MNase/µg of chromatin	None	None	None	None	None	None	None
**Prot. K treatment**	None	40 ng/µL for 5–10 min	None	None	None	None	None	80 ng/μL for 10 min at 40 °C
**Heating (LRC)**	None	None	1 h at 80 °C with 0.05% Tween-20	16 h at 65 °C with 0.5% Triton-X 100 and 0.1% sodium deoxycholate	16 h at 65 °C with 1% SDS	60 min at 65 °C followed by 30 min at 90 °C in 0.1% SDS	1 h at 50 °C in “ChIP buffer”	None
**Lysis (II)**	None	None	None	10 min in ice with 0.5% IGEPAL	None	None	None	None
**Sonication (paraffin emulsification)**	None	None	None	None	None	None	None	Covaris M220, 5 min at 20 °C (duty factor 20%, peak incident 75 W, 200 cycles per burst)
**Sonication (chromatin extraction)**	Probe sonicator, 3 × 30 s at 85% amplitude in 0.1% SDS	Covaris E210, 40 min in 0.1% SDS (duty factor 20%, intensity 8, 200 cycles per burst)	Probe sonicator, 3 × 30 s at 40% amplitude in 0.1% SDS	Bioruptor Twin (UCD-400), 3 × (30 × 30 s) in 1% Triton X-100, 0.1% sodium deoxycholate, 0.05% SDS	Covaris E220 5 min in 1% SDS (duty factor 5%, peak incident 105 W, 200 cycles per burst)	Bioruptor II, 60 × 30 s in 1% Triton-X 100, 0.5% IGEPAL (BM Equipment)	Probe sonicator, 40 × 30 s at 42% amplitude in “ChIP buffer”	Covaris M220, 10–30 min at 7 °C (duty factor 15%, peak incident 75 W, 200 cycles per burst) after the addition of “shearing buffer”

Conditions are reported only if published or described in kit datasheets. Buffer composition: in the absence of other relevant features, only the concentrations of detergents and salts are reported.

**Table 4 ijms-23-01103-t004:** Summary of the procedural steps used by the different FFPE-ChIP procedures for chromatin immunoselection and DNA purification.

	PAT-ChIP	FiT-seq	EPAT-ChIP	Chrom-EX PE	FiTAc-seq	RCRA ChIP-seq
**Immunoselection: input chromatin tested**	0.5–4 µg	N.A.	0.4–2 µg	N.A.	>0.4 µg (in the presence of 20 μg recombinant H2B and 1 μg reference total RNA)	0.4–2 µg
**Immunoselection: conditions**	16 h at +4 °C in 0.02% SDS, 50 mM NaCl	Protein G Agarose Columns (Active Motif)	16 h at +4 °C in 0.02% SDS, 50 mM NaCl	16 h in 1% Triton X-100, 0.1% sodium deoxycholate, 0.05% SDS	16 h at 4 °C in 0.1% SDS, 1% Triton X-100, 150 mM NaCl	In an ultrasonic water bath for 40 min in 150 mM NaCl, 1% Triton X-100, 0.5% IGEPAL
**Chromatin-ab complex separation**	Protein G-Sepharose beads		Protein G-Sepharose beads	Protein G magnetic beads	Protein A and protein G magnetic beads	Protein G magnetic beads
**Washes**	3 times with high volumes (10 mL) of 1% Triton X-100 and increasing NaCl concentrations (50, 100, and 150 mM)		3 times with high volumes (10 mL) of 1% Triton-X 100 and increasing NaCl concentrations (50, 100, and 150 mM)	4 times with: (i) 100 mM NaCl, 1% Triton X-100, 0.1% sodium deoxycholate; (ii) 500 mM NaCl, 1% Triton X-100, 0.1% sodium deoxycholate; (iii) 0.25 M LiCl2, 0.5% NP-40, 0.5% sodium deoxycholate; (iv) TE buffer	6 times with 0.7% sodium deoxycholate, 1% NP40, 0.5 M LiCl	4 times with: (i) 150 mM NaCl, 1% Triton X-100, 0.5% IGEPAL (2 times); (ii) 300 mM NaCl, 1% Triton X-100, 0.1% SDS, 0.1% Na-deoxycholate; (iii) 250 mM LiCl, 1% Triton X-100, 0.5% Na-deoxycholate
**Elution**	With 1% SDS		With 1% SDS	N.A.	With 1% SDS	With 1% SDS
**Decrosslinking**	16 h at 65 °C in the presence of 0.2 M NaCl	N.A.	16 h at 65 °C in the presence of 0.2 M NaCl	16 h at 65 °C	6–16 h at 65 °C	16 h at 65 °C in the presence of 0.2 M NaCl
**DNA purification**	Phenol–chloroform	N.A.	QIAquick PCR Purification Kit (Qiagen, Hilden, Germany)	MinElute PCR purification kit (Qiagen, Hilden, Germany)	QIAquick PCR Purification Kit (Qiagen, Hilden, Germany)	QIAquick PCR Purification Kit (Qiagen, Hilden, Germany) or Agencourt AMPure XP (Beckman Coulter, Brea, CA, USA)
**Repair**	None	None	None	None	None	PreCR Repair Mix (New England Biolabs, Beverly, MA, USA)

N.A. indicates that the information is not available in published manuscripts. Conditions are reported only if published or described in kit datasheets. Buffer composition: in the absence of other relevant features, only the concentrations of detergents and salts are reported.

## Data Availability

Not applicable.

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
