# Peer review of "The Current State of Chromatin Immunoprecipitation (ChIP) from FFPE Tissues"

_ijms, 2022, doi:10.3390/ijms23031103_

Round 1
Reviewer 1 Report
The manuscript, which title is “The current state of chromatin immunoprecipitation (ChIP) from FFPE tissues”, is interesting. And there are some questions in the manuscript.
- The relationship between the type of sample (such as tissue, animal, blood, culture cells) and treatment duration should add in the manuscript.
- The authors should add the advantage and disadvantage of differential methods. And the authors should list the relative references.
- The authors should provide the limitation of differential methods
Author Response
Reviewer 1
1 - The relationship between the type of sample (such as tissue, animal, blood, culture cells) and treatment duration should add in the manuscript.
As described in the manuscript, the application of a partial reversal of crosslinking by heat is a common feature of the most recent ChIP-FFPE techniques and seems to be fundamental to isolate chromatin of high quality from archival FFPE samples. If we have interpreted the point correctly, the Reviewer refers to the duration of this “treatment” in different types of sample. However, heating is not necessary when single-cells samples are used (e.g. blood, culture cells). The techniques described in our manuscript have been developed to work specifically with FFPE tissues and this is the reason why we didn’t focus the attention on their application to other kinds of samples (for which standard ChIP methods can be used). We hope the Reviewer will agree with our point of view.
2 - The authors should add the advantage and disadvantage of differential methods. And the authors should list the relative references.
We thank the Reviewer for this point. We discussed in the manuscript advantages and disadvantages that were reported by the same authors of the different methods, as well as those that we hypothesize basing on our experience. In many cases, the systematic assessment of these aspects is not easy, since different methods share common features, while others lack clear pros and cons. However, as suggested by the Reviewer, we checked the manuscript and made minor changes to emphasize, when possible, the reported advantages and disadvantages. Thank to this analysis, we also found that the main advantage of FACT-seq technique, that is the possibility to study low amounts of tissue, was not explicitly reported and we changed the text in function of this observation.
3 - The authors should provide the limitation of differential methods.
We thank the Reviewer for this point. Even in this case, all the limits described by the developers of the techniques or hypothesize by us have been already reported in the manuscript. Similarly to what we have done for point 2, we checked the manuscript and made minor changes to emphasize, when possible, the limits of the different procedures described.
Reviewer 2 Report
Amatori and Fanelli reviewed reports on chromatin immunoprecipitation (ChIP) methods using formalin fixed and paraffin embedded (FFPE) tissue samples. Properties, limitations, and pros and cons of each method are appropriately described. This manuscript is interesting, useful, and informative for a lot of researchers studying genome functions. Therefore, I recommend publishing this manuscript after minor revision.
Comments
1) Lines 54-65: References should be appropriately cited to describe history of development of standard ChIP methods.
2) Table 1: Indication of reference numbers would be useful for readers.
3) Table 1: Description of properties of FACT-Seq would be useful for readers.
4) Table 1: What does “N.A.” mean? Annotation (description but not simple definition) would be useful for readers.
5) Line 125: “locus-specific studies” may be difficult for readers to imagine. Does it mean studies using target-specific PCR or qPCR?
6) Lines 131-133: It would be informative if the authors could describe the reason why Fit-Seq failed to detect H3K27ac after these lines.
7) Table 2: “tissue preparation” in the title may be not suitable words.
8) Table 2: “Rnase A” is “RNase A”.
9) Table 3: Table 3 should be shown in one page, but not separated over two pages.
10) Line 409: “so- nicators” is “sonicators”.
11) Table 4: Table 4 should be shown in one page, but not separated over two pages.
12) Table 4: What does “N.A.” mean? Annotation (description but not simple definition) would be useful for readers.
13) Figure 3: As to the figure legend, “VCL” would be “ACTB”.
14) Lines 534-536: This sentence would be suitable in the paragraph, lines 484-494.
15) Lines 547-584: As to "2.5. A novel enzyme-tethering strategy”, insertion of a scheme like Figure 2 would be useful for readers.
16) Lines 588-589: Is it possible to add a link (web site) on “Roadmap Epigenomics Mapping Consortium”?
Author Response
Reviewer 2
1 - Lines 54-65: References should be appropriately cited to describe history of development of standard ChIP methods.
We thank the Reviewer for this point. Even if our intention was not to describe the history of ChIP development, we followed the Reviewer suggestion adding few main references regarding standard ChIP.
2 - Table 1: Indication of reference numbers would be useful for readers.
In accordance with Reviewer suggestion, reference numbers were added in Table 1.
3 - Table 1: Description of properties of FACT-Seq would be useful for readers.
Following the Reviewer suggestions, FACT-seq was added in Table 1.
4 - Table 1: What does “N.A.” mean? Annotation (description but not simple definition) would be useful for readers.
Following the Reviewer suggestion, we indicated what “N.A.” means in Table 1 footnotes.
5 - Line 125: “locus-specific studies” may be difficult for readers to imagine. Does it mean studies using target-specific PCR or qPCR?
Yes, the definition “locus-specific studies” refers to studies using target-specific PCR or qPCR. Following the Reviewer suggestion, we specified this information in line 125.
6 - Lines 131-133: It would be informative if the authors could describe the reason why Fit-Seq failed to detect H3K27ac after these lines.
We thank the Reviewer for this point. We described the possible reason why FiT-seq can’t be applied to H3K27ac in paragraph 2.2 with this sentence “…protein digestion may be “dangerous” considering that a protein is just the target of the assay. The failure in the application of the FiT-seq by our research group and the statement by the authors themselves that FiT-seq fails to work with H3K27ac may indicate the necessity of a tricky fine-tuning of the technique”. However, the developers of the technique did not explain the reason of this failure in their manuscripts and, in the absence of experimental evidence, the disruption of H3K27ac by proteinase K digestion is just a reasonable hypothesis. In this regard, we would like to underline that the manuscript describing FiT-seq was published in Nature Medicine in 2016 with poor methodological details and in 2019 a second paper describing H3K27ac and H3K4me2 investigation “with modified buffer and sonication conditions” of the original method was published in the same journal. It is not possible to know for sure which method has been used for the two immunoprecipitations against H3K27ac and H3K4me2, even if it seems that the same method has been used. The year after, the paper describing in detail a new method specific for H3K27ac, named FiTAc-seq, was published by the same authors in Nature Protocols. As reported in our review, the main novelty of this method, respect to the previous, is the elimination of proteinase K digestion and the use of heat to partially reverse the crosslinking of FFPE tissues, a strategy already introduced by two other methods (EPAT-ChIP - 2018, and Chrom-EX PE - 2019). In the Nature Protocols paper the authors declared that their new procedure was already applied in the 2019 manuscript in which they studied also H3K4me2 and in which we did not found references to the usage of high temperatures.
Finally, it has also to be considered that the possibility that proteinase K treatment may interfere with other epitopes in addition to H3K27ac is, in our opinion, extremely high.
Thus, following the Reviewer suggestion, we change the sentence at lines 131-133 from “suggesting a lack of compatibility with histone acetylation” to “suggesting a possible degradation of H3K27ac due to proteinase K activity that may potentially affect also other kinds of epitope in certain experimental conditions”.
7 - Table 2: “tissue preparation” in the title may be not suitable words.
Following the Reviewer suggestion, we changed “tissue preparation” with “tissue processing before chromatin isolation”.
8 - Table 2: “Rnase A” is “RNase A”.
We changed the typing error.
9 - Table 3: Table 3 should be shown in one page, but not separated over two pages.
We thank the Reviewer for this point. However, the only way to show Table 3 in one page without losing fundamental information is to reduce the font size with a consequent impact on table readability. We will expect to get instructions from the editorial team of the journal to find the best solution to overcome this problem.
10 - Line 409: “so- nicators” is “sonicators”.
We changed the typing error.
11 - Table 4: Table 4 should be shown in one page, but not separated over two pages.
Please, see reply to point 9.
12 - Table 4: What does “N.A.” mean? Annotation (description but not simple definition) would be useful for readers.
Following the Reviewer suggestion, we indicated what “N.A.” means in Table 4 notes.
13 - Figure 3: As to the figure legend, “VCL” would be “ACTB”.
Following the Reviewer suggestion, we changed “VCL” with “ACTB” in Figure 3 legend.
14 - Lines 534-536: This sentence would be suitable in the paragraph, lines 484-494.
Following the Reviewer suggestion, we added this sentence in the paragraph lines 484-494: “In addition, coupling this test with the real-time qPCR analysis of regions known to be enriched or not enriched by the histone PTM under study, can provide an idea of the enrichment that can be reached with a specific immunoglobulin (see subsequent paragraph for details)”.
15 - Lines 547-584: As to "2.5. A novel enzyme-tethering strategy”, insertion of a scheme like Figure 2 would be useful for readers.
We thank the Reviewer for this point. However, although we consider FACT-seq technique important, as it represents the first attempt to apply enzyme-tethering strategies to FFPE tissues, we preferred to focus the attention on the techniques based on chromatin isolation, for which we have a high level of experience due to several years of work in the field and the publication of two original methods. On the contrary, although we consider a duty to cite this procedure, the lack of a deep knowledge about enzyme-tethering strategies prompts us to do not dedicate a specific figure to this procedure, also considering that the manuscript already contains 3 figures and 4 extended tables and that a scheme of the method is already reported in the original FACT-seq publication. However, in case the Reviewer will not agree with our point of view, we can consider anyway the adding of a new figure.
16 - Lines 588-589: Is it possible to add a link (web site) on “Roadmap Epigenomics Mapping Consortium”?
As suggested by the Reviewer, we added the web site of the Roadmap Epigenomics Mapping Consortium”.
Round 2
Reviewer 1 Report
No more question.